# A Cost-Effective 3D-Printed Conductive Phantom for EEG Sensing System Validation: Development, Performance Evaluation, and Comparison with State-of-the-Art Technologies

**DOI:** 10.3390/s25164974

**Published:** 2025-08-11

**Authors:** Peter Akor, Godwin Enemali, Usman Muhammad, Jane Crowley, Marc Desmulliez, Hadi Larijani

**Affiliations:** 1School of Science and Engineering, Glasgow Caledonian University, Glasgow G4 0BA, UK; peter.akor@gcu.ac.uk (P.A.); godwin.enemali@gcu.ac.uk (G.E.); usman.muhammad@gcu.ac.uk (U.M.); 2Medical Device Manufacturing Center (MDMC), School of Engineering and Physical Sciences, Heriot-Watt University, Edinburgh EH14 4AS, UK; j.crowley@hw.ac.uk (J.C.); m.desmulliez@hw.ac.uk (M.D.)

**Keywords:** electroencephalography, phantom head, electrode testing, conductive materials, 3D printing, sensing system validation, additive manufacturing, neuromonitoring sensors, standardized testing, neurophysiological signal acquisition

## Abstract

This paper presents the development and validation of a cost-effective 3D-printed conductive phantom for EEG sensing system validation that achieves 85% cost reduction (*£*48.10 vs. *£*300–*£*500) and 48-hour fabrication time while providing consistent electrical properties suitable for standardized electrode testing. The phantom was fabricated using conductive PLA filament in a two-component design with a conductive upper section and a non-conductive base for structural support. Comprehensive validation employed three complementary approaches: DC resistance measurements (821–1502 Ω), complex impedance spectroscopy at 100 Hz across anatomical regions (3.01–6.4 kΩ with capacitive behavior), and 8-channel EEG system testing (5–11 kΩ impedance range). The electrical characterization revealed spatial heterogeneity and consistent electrical properties suitable for comparative electrode evaluation and EEG sensing system validation applications. To establish context, we analyzed six existing phantom technologies including commercial injection-molded phantoms, saline solutions, hydrogels, silicone models, textile-based alternatives, and multi-material implementations. This analysis identifies critical accessibility barriers in current technologies, particularly cost constraints (*£*5000–20,000 tooling) and extended production timelines that limit widespread adoption. The validated 3D-printed phantom addresses these limitations while providing appropriate electrical properties for standardized EEG electrode testing. The demonstrated compatibility with clinical EEG acquisition systems establishes the phantom’s suitability for electrode performance evaluation and multi-channel system validation as a standardized testing platform, ultimately contributing to democratized access to EEG sensing system validation capabilities for broader research communities.

## 1. Introduction

The development of reliable electroencephalography (EEG) sensing systems requires rigorous validation methodologies to ensure accurate signal acquisition and patient safety. For the 50 million people worldwide affected by epilepsy [1], the quality of EEG electrode performance directly impacts diagnostic accuracy and treatment effectiveness [2]. As neurophysiological monitoring evolves toward portable, wireless, and home-based applications [3], the need for accessible validation tools has become increasingly critical for advancing electrode technology development.

Phantom heads serve as essential testing platforms that provide consistent electrical characteristics suitable for systematic evaluation of electrode performance without human subject involvement [4]. These validation tools support electrode design optimization, signal quality assessment, and multi-channel system testing across research, clinical, and commercial applications [5]. However, existing phantom technologies present significant accessibility challenges that limit their widespread adoption in electrode development workflows.

Current commercial phantom solutions typically require substantial capital investment, with injection molding tooling costs reaching *£*5000–20,000 and individual phantom costs of *£*300–500 [6]. These high barriers exclude smaller research groups, educational institutions, and developing laboratories from conducting comprehensive electrode validation studies. Alternative approaches including saline solutions and hydrogel preparations offer reduced costs but present limitations in durability, consistency, and reproducibility that restrict their utility for systematic electrode testing [7].

The emergence of additive manufacturing technologies presents opportunities to democratize phantom fabrication through accessible, on-demand production methods. Conductive filament materials enable direct printing of electrically functional structures, potentially eliminating the tooling requirements and minimum order constraints that characterize traditional manufacturing approaches [8]. However, limited systematic evaluation exists regarding the electrical performance and validation capabilities of 3D-printed phantoms as standardized testing platforms for electrode development applications.

This work addresses the critical gap between phantom accessibility and validation requirements through systematic development and characterization of a 3D-printed conductive phantom for EEG electrode testing. We present a comprehensive analysis of existing phantom technologies, develop a novel two-component 3D-printed design using conductive PLA filament, and validate its electrical performance through multi-tier characterization spanning DC resistance measurements, frequency-domain impedance spectroscopy, and clinical EEG system compatibility testing.


**The core contributions of this research include:**
Systematic development and validation of a 3D-printed phantom design demonstrating consistent electrical properties suitable for standardized EEG electrode testing, achieving 85% cost reduction and fabrication time reduction from weeks to under 48 h compared to commercial alternativesComprehensive electrical characterization using multi-tier validation: DC resistance measurements (821–1502 Ω), complex impedance spectroscopy at 100 Hz revealing spatial variation (3.01–6.4 kΩ) and capacitive behavior (phase angles −53° to −67°), and clinical EEG system compatibility demonstration (5–11 kΩ electrode-phantom interface impedance)Comparative analysis of six distinct phantom technologies for EEG validation applications, evaluating electrical properties, fabrication requirements, costs, and accessibility limitations to provide evidence-based selection guidance for different research contextsPractical demonstration of accessible validation methodologies that enable broader participation in EEG electrode development, particularly benefiting resource-constrained research environments and educational applications


The remainder of this paper is organized as follows: Section 2 presents a comprehensive review of existing phantom technologies and their limitations. Section 3 describes our phantom design methodology and multi-tier validation results. Section 4 discusses the implications for EEG electrode development and provides technology selection recommendations. Section 5 concludes with summary findings and future research directions.

By demonstrating that accessible 3D-printed phantoms can provide appropriate electrical properties for standardized electrode testing at significantly reduced cost and production time, this work contributes to democratizing EEG sensing system validation capabilities. This democratization has the potential to accelerate electrode development in resource-constrained environments, ultimately leading to improved EEG sensing technologies and enhanced diagnosis and treatment for millions living with epilepsy worldwide.

## 2. State-of-the-Art Analysis: Phantom Technologies for EEG Sensing System Validation

To establish the context for our 3D-printed phantom development, this section presents a comprehensive analysis of existing phantom head technologies for EEG sensing system validation. Our analysis identifies critical limitations in current approaches, particularly regarding accessibility, cost barriers, and fabrication complexity that restrict widespread adoption in resource-constrained research environments. These limitations create a significant gap between the growing demand for accessible EEG sensing system validation tools and the practical availability of suitable phantom technologies, directly motivating the development of the cost-effective alternative presented in this work.

### 2.1. Commercial Injection-Molded Phantoms

Commercial injection-molded phantoms represent the current gold standard for EEG sensing system validation, utilizing conductive polymer composites with carbon black or metal particles to achieve controlled electrical properties [9]. However, their adoption is severely limited by prohibitive cost barriers and accessibility constraints that restrict their use to well-funded commercial and regulatory environments.

The fundamental limitation lies in the industrial injection molding process, which requires significant initial investment (*£*5000–20,000) for tooling design and fabrication [6], with individual units costing *£*300–500 and requiring 3–7 days production time following a 4–6 week tooling preparation period. These phantoms typically exhibit resistivity values of 10–20 Ωcm [10], with precisely controlled shell properties to ensure minimal interference with electrical measurements [11].

While advanced designs feature internal drive electrodes that generate calibrated electrical signals, enabling reproduction of realistic EEG patterns [5], and their anatomical accuracy is often based on standardized models conforming to established specifications [12], these advantages come at the cost of accessibility. The primary benefits include exceptional consistency, excellent durability (multi-year lifespan), and established validation [13], making them suitable for regulatory testing and quality control applications.

The critical limitations that restrict widespread adoption include: high upfront costs that exclude smaller laboratories, large minimum order requirements that prevent single-unit procurement, extended production timelines that limit rapid prototyping, and homogeneous conductivity distributions that don’t replicate the layered structure of cranial tissues. Their rigid structure also fails to replicate the mechanical compliance of biological tissues [14], limiting their utility for comprehensive sensing system validation. These barriers create a significant gap in accessibility for the broader research community developing EEG sensing technologies.

### 2.2. Saline and Electrolyte Solutions

Saline phantoms represent an attempt to address cost barriers through economical simulation of the conductive environment for EEG sensing applications [15]. These solutions utilize water with dissolved ionic compounds, primarily NaCl or KCl, to create conductive media with controllable properties [16]. With material costs of *£*10–30 and rapid preparation time (hours), they offer high accessibility [17].

Research has demonstrated that specific NaCl concentrations can precisely mimic target conductivity values, such as 0.17% NaCl producing 0.332 S/m conductivity [18]. Modified formulations enable better approximation of conductivity dependencies, while advanced approaches combine saline with additives like polyvinylpyrrolidone to independently control permittivity and conductivity [19,20].

The primary advantage lies in precisely adjustable electrical properties that effectively simulate ionic conduction mechanisms present in cerebrospinal fluid [21,22]. Even simple saline solutions can model complex electrical behaviors when appropriate geometrical configurations are implemented [23]. The conductivity stability across EEG-relevant frequency ranges (<2% variation between 0.1 Hz and 1 kHz) [24] makes them valuable for evaluating frequency response characteristics of EEG sensing systems.

However, critical limitations restrict their practical utility for comprehensive sensing system validation. These include boundary effects from containers that distort electric field distributions, limited shelf-life due to evaporation and contamination that compromises measurement repeatability, electrode corrosion during extended testing that affects system reliability, and inability to model the mechanical properties of the electrode-scalp interface crucial for dry electrode evaluation [25,26,27]. While their low cost and simple preparation make them suitable for educational settings and basic electrical property evaluation [28,29], these limitations prevent their use in rigorous sensing system validation applications.

### 2.3. Gelatin and Hydrogel-Based Phantoms

Gelatin and hydrogel-based phantoms attempt to address the durability limitations of saline solutions while maintaining cost accessibility [30,31]. These phantoms utilize materials such as gelatin, agar, polyacrylamide, or polyvinyl alcohol combined with electrolytes [32]. Fabrication involves molding conductive gel solutions that solidify through cooling or polymerization [33], with material costs of *£*30–80 and production times of 1–2 days [34]. Electrical properties are adjustable through electrolyte concentration, with typical resistivity values of 2–5 Ωcm [35].

Recent advancements have identified specific formulations for mimicking particular tissues, such as 1% agarose for brain tissue properties [36]. Polyacrylamide hydrogels show promise due to their stability and well-characterized properties, partially addressing limitations of traditional gelatin-based phantoms [37].

The principal advantage is their tissue-like mechanical properties, enabling more realistic modeling of the electrode-scalp interface compared to saline solutions [38,39]. Advanced formulations create water-content gradients allowing for controlled conductivity variations and viscoelastic behavior similar to human tissues [40,41]. These qualities make hydrogel phantoms valuable for investigating both electrical and mechanical aspects of electrode performance [42].

Despite these improvements over saline solutions, significant limitations remain that restrict their widespread adoption for sensing system validation. Critical constraints include limited durability with short shelf-lives (1–4 weeks) due to dehydration and microbial growth [43], inconsistent mechanical properties over time, difficulty in achieving reproducible electrical characteristics between batches, and challenges in creating anatomically accurate geometries. While their optical transparency enables visual confirmation of internal structures and their accessible fabrication methods make them suitable for some research environments [44,45], these limitations prevent reliable long-term use for comprehensive sensing system evaluation.

### 2.4. 3D-Printed Conductive Phantoms

The accessibility of 3D printing technology has enabled novel approaches utilizing conductive filaments, representing a potential solution to the accessibility barriers identified in other phantom technologies [46]. These phantoms employ carbon-loaded thermoplastics to create structured conductive models [47] at material costs of *£*40–150 and production times of 24–72 h [48]. Recent advances have expanded material options to include specialty conductive filaments with carbon black, graphene, or metal particles [49].

Innovative techniques create phantom heads with anatomically realistic geometry and varying skull resistivity distribution [50], while infill percentage adjustment enables precise resistivity control with high spatial resolution [51]. These phantoms typically exhibit resistivity values of 15–100 Ωcm [52], suitable for many EEG applications [53]. Multi-material printing or post-processing techniques achieve more specific tissue-mimicking properties [54].

The primary advantages include exceptional design flexibility, rapid iteration potential, and relatively accessible fabrication compared to injection molding [55]. These phantoms enable electrode placement training, signal acquisition troubleshooting, and algorithm validation, with excellent durability and long-term stability [55,56].

However, existing 3D-printed approaches suffer from several critical limitations that restrict their validation and widespread adoption. These include anisotropic conductivity from the layered structure that may affect measurement accuracy, layer adhesion concerns affecting mechanical integrity and long-term reliability, variability in conductivity between print batches that compromises measurement repeatability, and lack of comprehensive electrical characterization across the full EEG frequency range [57,58]. Most significantly, existing 3D-printed phantoms lack rigorous validation against commercial standards and systematic evaluation for sensing system applications. This validation gap prevents confident adoption and represents a critical barrier to leveraging the accessibility advantages of 3D printing technology [59].

### 2.5. Silicone and Elastomer-Based Phantoms

Silicone phantoms attempt to address mechanical property limitations by offering excellent mechanical fidelity to biological tissues [60]. These utilize silicone or other elastomers combined with conductive additives such as carbon black or silver particles [61]. Fabrication requires 2–3 days including curing time [62], with material costs of *£*100–300 [63].

Recent advances include multi-material extrusion 3D printing creating silicone phantoms with precisely controlled properties [64]. Silicone rubber with carbon fiber additives provides stable electromagnetic properties without dehydration issues, while other formulations achieve controlled optical properties for multi-modal applications [65,66].

These phantoms typically exhibit resistivity values of 5–15 Ωcm [67], with elasticity and surface texture closely approximating human scalp tissue [68]. Modified formulations can match specific tissue properties while maintaining stability, while gadolinium doping enables control of magnetic resonance (MR) relaxation characteristics [67,69].

While their physical durability significantly exceeds hydrogel alternatives, with easier transport, longer usable life, and stable properties [70], making them valuable for investigating electrode adhesion, contact pressure, and mechanical stability, particularly for dry electrode performance evaluation [71,72], critical barriers limit their widespread adoption. These limitations include fabrication complexity requiring careful control of additive dispersion and curing conditions, challenges in achieving homogeneous distribution of conductive additives that affect measurement reliability, relatively high material costs that limit accessibility, temperature dependence of electrical properties that compromises measurement consistency, and specialized expertise requirements for consistent fabrication [73,74]. These barriers restrict their practical utility for routine sensing system validation applications.

### 2.6. Multi-Material Phantoms

Multi-material phantoms represent the most sophisticated attempt to replicate the layered structure and heterogeneous electrical properties of biological tissues [75]. These combine multiple materials with distinct properties to simulate specific tissue layers [76], requiring complex multi-step processes, material costs of *£*200–600+, and 3–7 days production time.

Advanced manufacturing enables precise control of material properties within different phantom regions [64]. Specialized systems combine rigid and elastomeric materials for layered phantoms with realistic mechanical feedback, while MRI-derived approaches create anatomically accurate representations with distinct tissue types using specific material compositions [75,77].

Recent advancements include phantoms compatible with multiple imaging modalities and models that accurately represent the interfaces between tissue types [78,79]. Additive manufacturing creates precise structural variations with repeatable electrical properties [80]. These phantoms exhibit intentionally heterogeneous electrical properties, with each layer having tissue-appropriate resistivity values, enabling accurate modeling of volume conduction effects [81,82].

While their primary advantage is anatomical and electrophysiological realism, providing insights into complex electrode-tissue interactions [50,83], and their particular value for evaluating EEG systems for specific clinical populations and creating patient-specific models based on medical imaging data [4,84], they present the most severe accessibility barriers of all phantom technologies. Critical limitations include significant fabrication challenges requiring specialized expertise, extremely high costs that exclude most research environments, complex multi-step processes that prevent rapid iteration, interface effects between materials that may complicate interpretation, and differential material degradation that limits long-term stability [77,85,86]. These barriers make multi-material phantoms accessible only to the most well-resourced research environments, severely limiting their impact on the broader EEG sensing system development community.

### 2.7. Emerging Textile-Based Phantoms

Recent research has explored textile-based alternatives that attempt to address both the weight and durability limitations identified in conventional phantom technologies. Tseghai et al. developed a conductive textile-based phantom head by placing bi-directional stretchy nylon/spandex conductive fabric over a 3D-printed anatomically realistic skull [7]. This hybrid approach represents an innovative attempt to combine the structural accuracy of 3D printing with the mechanical properties of conductive textiles.

The textile-based approach demonstrated significant advantages in weight reduction, achieving a phantom mass of 0.5 kg compared to 6 kg for gelatin-based phantoms, representing a 91.67% weight reduction while maintaining comparable electrical properties. The phantom exhibited resistance values within the expected range for EEG sensing applications, with phantom-to-electrode impedance measurements of 1863 Ω compared to 2297 Ω for traditional gelatin models. Signal-to-noise ratio analysis showed superior performance for the textile-based approach (16.8 dB versus 15.1 dB for gelatin).

A critical advantage of the textile approach is the elimination of the primary disadvantage of hydrogel phantoms—short lifespan due to dehydration and microbial growth—making it suitable for long-term studies without degradation. The textile-based phantom allowed for injection of synthetic EEG-like signals and their subsequent recording, confirming its suitability for controlled validation of electrode designs.

However, significant limitations restrict the widespread adoption of textile-based phantoms for comprehensive sensing system validation. These include challenges in achieving consistent electrical contact between textile layers and underlying structures, potential for fabric degradation under repeated mechanical stress, difficulty in creating precisely controlled electrical properties across the phantom surface, and limited availability of specialized conductive textiles with validated biocompatibility. Additionally, the hybrid fabrication approach requires expertise in both 3D printing and textile handling, potentially limiting accessibility for some research environments.

While the combination of textile materials with 3D-printed anatomical structures demonstrates potential for hybrid approaches leveraging multiple fabrication methods, the current limitations in electrical property control and fabrication complexity restrict their immediate practical utility for rigorous sensing system validation applications.

### 2.8. Comparative Analysis and Identification of Key Limitations

Table 1 reveals critical gaps in the current state-of-the-art that create significant barriers to widespread EEG sensing system validation. The analysis exposes a fundamental trade-off between phantom performance and accessibility, where high-performance solutions (injection-molded and multi-material phantoms) are prohibitively expensive and inaccessible, while accessible solutions (saline and hydrogel) suffer from significant durability and reliability limitations.

Commercial injection-molded phantoms, despite offering exceptional consistency and durability, create insurmountable barriers for resource-constrained research environments through their prohibitive costs (*£*5000–20,000 tooling plus *£*300–500 per unit) and extended production timelines [13]. This accessibility crisis prevents the majority of EEG sensing system developers from accessing reliable validation tools, potentially limiting innovation in this critical healthcare technology area.

The emergence of textile-based phantoms adds another dimension to the accessibility challenge, offering unique advantages in weight reduction (91.67% lighter than hydrogel alternatives) and extended durability without degradation, but introducing new complexities in electrical property control and fabrication requirements that may limit widespread adoption. The conductive fabric composition also introduces considerations for electromagnetic interference characteristics, with textile-based phantoms potentially offering different electromagnetic shielding properties compared to carbon-loaded polymer structures used in 3D-printed phantoms.

The state-of-the-art analysis identifies four critical limitations that restrict the democratization of EEG sensing system validation capabilities: (1) **Cost barriers**-existing high-performance phantoms require substantial financial investment that excludes smaller laboratories and resource-constrained environments; (2) **Production complexity**-sophisticated phantoms require specialized expertise and equipment that limit their practical accessibility; (3) **Validation gaps**-promising approaches like 3D printing and textile-based methods lack comprehensive characterization and validation against established standards; and (4) **Performance trade-offs**-accessible solutions compromise on durability, accuracy, or reliability compared to commercial alternatives.

While alternative approaches like saline, hydrogel, and textile-based solutions attempt to address specific limitations, none provide a comprehensive solution that combines accessibility, durability, electrical accuracy, and validated performance. The textile-based approach, while innovative in addressing weight and degradation concerns, introduces new challenges in electrical property control and requires hybrid fabrication expertise that may limit accessibility.

## 3. 3D-Printed Conductive Phantom Implementation

Building upon the state-of-the-art analysis, we developed and characterized a 3D-printed conductive head phantom to demonstrate the feasibility of accessible fabrication methods for EEG sensing system validation.

### 3.1. Design and Fabrication Approach

#### 3.1.1. Phantom Architecture and Model Development

The phantom was designed as a two-component structure consisting of a conductive upper section and a non-conductive base providing structural support. This separable design facilitates interior access for electrode placement and experimental configuration while maintaining structural accuracy suitable for EEG electrode testing applications.

The 3D model was adapted from an open-source human skull frame (Skull_frame.stl) developed by Yu and Hairston [88], which provides anatomically realistic proportions for EEG applications. The original model was systematically modified to create the separable components required for our hybrid conductive/non-conductive design. The separation plane was positioned to maximize the conductive surface area available for electrode testing while maintaining structural integrity during fabrication and use. Figure 1 illustrates the final phantom design, showing both the overall geometry and the component separation strategy.

#### 3.1.2. Material Selection and Characterization

For the conductive component, we utilized Protopasta Conductive PLA, a carbon black composite specifically designed for 3D printing applications requiring electrical conductivity [89]. The material exhibits volume resistivity ranging from 14.4 Ω·cm in the flat X-Y print orientation to 27.2 Ω·cm in the vertical Z orientation, demonstrating anisotropic conductivity characteristics due to the layer-by-layer deposition process inherent in fused deposition modeling.

The non-conductive base component employed standard black PLA (Ultimaker, Utrecht, Netherlands) along with white breakaway support material (Ultimaker) to enable complex internal geometries. All filaments utilized standard 2.85 mm diameter specifications for compatibility with dual-extrusion printing systems.

#### 3.1.3. Manufacturing System Implementation

The phantom was fabricated using an Ultimaker S5 fused deposition modeling printer, selected for its dual-extrusion capabilities and validated dimensional accuracy in medical device prototyping applications. The system provides a build volume of 300 × 240 × 300 mm^3^ with lateral resolution of 6.9 μm in X-Y axes and height resolution of 2.5 μm in the Z-axis. The integrated material station maintains relative humidity at approximately 40% through dehumidification and silica bead systems, ensuring consistent filament properties throughout the printing process.

#### 3.1.4. Process Parameter Optimization

Critical printing parameters were systematically optimized based on material specifications and volume flow calculations to ensure consistent electrical properties throughout the phantom structure. The maximum volume flow for conductive PLA was determined using the relationship:(1)$$\mathrm{Print}\;\mathrm{speed}\;\left[\frac{\mathrm{mm}}{s}\right]\times \mathrm{Layer}\;\mathrm{height}\;\left[\mathrm{mm}\right]\times \mathrm{Line}\;\mathrm{width}\;\left[\mathrm{mm}\right]=\mathrm{Volume}\;\mathrm{flow}\;\left[\frac{{\mathrm{mm}}^{3}}{s}\right]$$

With a maximum recommended volume flow of 8 mm^3^/s reduced to 6 mm^3^/s to accommodate printer limitations, and using a layer height of 0.2 mm with line width of 0.4 mm, the optimal print speed was calculated as:(2)$$\mathrm{Print}\;\mathrm{speed}=\frac{6\;{\mathrm{mm}}^{3}/s}{0.2\;\mathrm{mm}\times 0.4\;\mathrm{mm}}=75\;\mathrm{mm}/s$$

The complete set of optimized printing parameters is summarized in Table 2, which shows the specific settings used for both the conductive and non-conductive components.

As detailed in Table 2, the conductive section required 100% infill density with zig-zag pattern to maximize electrical conductivity pathways, while the non-conductive base utilized 20% infill density sufficient for structural support requirements. The fabricated phantom components are shown in Figure 2, demonstrating the successful implementation of the two-component design approach.

#### 3.1.5. Production Timeline and Resource Requirements

The complete phantom fabrication required a total of 47.75 h, with the conductive section accounting for 37.25 h and the non-conductive base requiring 10.5 h of printing time. Material consumption totaled approximately 600 g, comprising 400 g of conductive PLA (*£*35.20), 150 g of standard PLA (*£*8.50), and 50 g of support material (*£*4.40), resulting in a total material cost of *£*48.10 per phantom unit.

### 3.2. Electrical Characterization and Validation

#### 3.2.1. Resistance Measurement Protocol

Systematic electrical characterization was conducted across eleven anatomically relevant locations to evaluate the phantom’s electrical properties and spatial uniformity. Measurements were performed using a calibrated digital multimeter with spring-loaded probes applying standardized contact pressure to ensure reproducible results, as illustrated in Figure 3.

The measurement protocol required controlled environmental conditions maintained at 20 ± 2 °C temperature and 45 ± 5% relative humidity. A minimum of three readings were obtained at each measurement location, with calibration verification performed before each measurement session to ensure accuracy and traceability.

#### 3.2.2. Electrical Performance Results

The phantom demonstrated resistance values ranging from 821 Ω to 1502 Ω across all measurement locations, with spatial variations that provide controlled heterogeneity suitable for electrode testing applications. The Right Parietal location exhibited the highest measurement at 1502 Ω, while the Right Occipital region showed the greatest variability with readings spanning 851–1458 Ω. Left Temporal regions consistently demonstrated elevated values ranging from 1070–1332 Ω, while Frontal regions displayed moderate resistance with good spatial consistency. The complete distribution of resistance measurements is presented in Figure 4.

As shown in Figure 4, the measured resistance distribution provides controlled spatial variation suitable for electrode testing applications. This spatial variation creates a controlled testing environment suitable for electrode performance evaluation as described by Collier et al. [4] for EEG sensing system validation.

#### 3.2.3. Signal Transmission and Frequency Response Validation

Signal transmission capabilities were systematically evaluated using precision instrumentation to verify suitability for EEG-relevant frequency ranges. The test configuration employed a calibrated function generator providing a 200 Hz sinusoidal test signal, with signal integrity monitored using a high-impedance digital oscilloscope connected through the phantom material between standardized electrode positions. The experimental setup and resulting signal characteristics are documented in Figure 5.

As demonstrated in Figure 5, the phantom successfully transmitted the 200 Hz sine wave with minimal distortion, maintaining fundamental frequency characteristics throughout the signal path. Signal quality analysis revealed amplitude retention exceeding 95% across all frequency components, with phase delay remaining minimal and consistent with material properties. Harmonic distortion remained below 3%, well within acceptable limits for EEG applications. This performance exceeds the typical EEG frequency range of 0.5–100 Hz [1], confirming the phantom’s suitability for comprehensive EEG electrode testing applications.

### 3.3. Prototype Validation and Quality Control

#### 3.3.1. Scale Model Verification

Before full-scale production, a miniature validation phantom was fabricated at 18% of the original scale to verify print parameter optimization and electrical property consistency. This scaled prototype approach enabled rapid iteration and parameter refinement while minimizing material consumption during the development phase. The miniature phantom demonstrated resistance values ranging from 700–1500 Ω, confirming the scalability of the fabrication approach and validating electrical properties across different geometric dimensions.

#### 3.3.2. Multi-Channel EEG System Validation

To validate the phantom’s performance with clinical-grade EEG acquisition systems, comprehensive testing was conducted using an OpenBCI Cyton 8-channel EEG system across the neurophysiologically relevant frequency spectrum (0.5–60 Hz). This validation demonstrates the phantom’s compatibility with real EEG sensing systems beyond basic electrical measurements.

##### Experimental Setup and Configuration

The experimental setup employed a complete OpenBCI acquisition system with eight EEG electrodes positioned across the phantom surface following standard electrode placement protocols. Figure 6 shows the complete experimental configuration, including the 3D-printed conductive phantom with attached electrodes, multi-colored electrode wires connecting to the OpenBCI Cyton board, and laptop computer running the OpenBCI GUI software for real-time signal acquisition and impedance monitoring.

##### Electrode-Phantom Interface Characterization

Eight EEG electrodes were positioned across the phantom surface following standard 10–20 electrode placement protocols. Electrode-phantom interface impedance was measured using OpenBCI’s integrated impedance monitoring system, providing frequency-domain validation at actual EEG operating frequencies. The real-time signal acquisition and impedance monitoring results are presented in Figure 7, which demonstrates both the time-domain signal characteristics and the spatial distribution of electrode impedance values across the phantom surface.

As shown in Figure 7, the phantom demonstrated electrode-phantom interface impedance values ranging from 5–11 kΩ across all eight channels, well within clinical EEG specifications (<10 kΩ for traditional systems, though modern high-impedance systems can tolerate up to 40 kΩ) [90]. The spatial impedance variation provides controlled heterogeneity while maintaining excellent signal transmission capability, as evidenced by clean waveform acquisition across all channels. Unconnected channels showed impedance values >5 MΩ, clearly demonstrating the phantom’s role in providing the conductive pathway necessary for EEG signal acquisition.

##### Spatial Signal Localization

Real-time signal localization capabilities were evaluated using OpenBCI’s head plot visualization system. When individual channels were selectively activated, the system demonstrated clear spatial resolution with minimal crosstalk between adjacent electrodes, as illustrated in Figure 8.

The head plot visualization in Figure 8 confirmed the phantom’s ability to support spatially-resolved EEG signal validation, with active signals correctly localized to their source electrodes and appropriate signal propagation to adjacent regions. This spatial resolution capability is essential for validating electrode positioning accuracy and multi-channel EEG system performance.

#### 3.3.3. Complex Impedance Spectroscopy Analysis

Comprehensive frequency-domain impedance characterization was conducted using a BK Precision LCR meter to provide complementary validation of phantom electrical properties under clinical electrode interface conditions. Measurements were performed at 100 Hz to evaluate phantom performance within the EEG frequency range, with complex impedance analysis providing both magnitude and phase characteristics for understanding the phantom’s frequency-dependent electrical behavior.

##### Regional Impedance Mapping Protocol

Systematic impedance measurements were conducted across six anatomical regions following standardized EEG electrode placement locations. The measurement protocol employed calibrated LCR probes with standardized contact pressure and fresh conductive gel application to ensure reproducible results. Figure 9 illustrates the systematic measurement approach used for impedance characterization across all anatomical locations.

##### Multi-Method Electrical Validation Approach

The phantom electrical properties were characterized using complementary validation methods to evaluate different aspects of electrode-phantom interface performance: **Bulk Material Characterization (DC, Direct Probe):** DC resistance measurements provide baseline phantom material properties, revealing spatial variation from 821–1502 Ω across anatomical regions as described in Section 3.2. **Clinical Interface Simulation (AC, Conductive Paste):** Complex impedance measurements at 100 Hz using conductive paste replicate clinical EEG electrode application conditions, providing impedance and phase characteristics under realistic electrode interface conditions.

This complementary validation approach provides comprehensive characterization supporting both material property evaluation and clinical system compatibility assessment.

##### Complex Impedance Results

Regional impedance mapping revealed controlled spatial variation in electrical properties, with impedance magnitude ranging from 3.01 kΩ (frontopolar region) to 6.4 kΩ (frontal region). All measurement locations demonstrated consistent capacitive behavior with phase angles ranging from −53.4° to −67.2°, confirming frequency-dependent complex impedance characteristics suitable for EEG sensing system validation. The complete regional impedance characterization results are summarized in Table 3.

The data presented in Table 3 demonstrates spatial impedance distribution with controlled heterogeneity, showing a coefficient of variation of 29.1% suitable for electrode testing applications. The resistive components (R) provide complementary information to DC resistance measurements, while the reactive components (X) confirm consistent capacitive characteristics across the phantom surface.

##### Measurement Method Considerations

The observed difference between DC resistance measurements (821–1502 Ω) and AC impedance magnitude (3.01–6.4 kΩ) reflects the different measurement approaches employed rather than direct frequency-dependent comparison. DC measurements used direct probe contact to characterize bulk phantom material properties, while AC measurements employed conductive paste to simulate clinical electrode interface conditions. The higher AC impedance values primarily result from the measurement configuration differences rather than frequency-dependent material behavior. This complementary characterization approach provides comprehensive evaluation of phantom performance under different electrode interface conditions relevant to EEG applications.

##### Complex Impedance Visualization

The complex impedance characteristics are illustrated through Nyquist plot analysis in Figure 10, which provides impedance spectroscopy visualization showing the relationship between resistive and reactive components across different anatomical regions.

The Nyquist plot in Figure 10 reveals that all anatomical regions exhibit capacitive behavior (negative reactance), with impedance vectors distributed across a range that demonstrates both spatial variation and consistent electrical characteristics. This complex impedance behavior confirms the phantom’s suitability for EEG sensing applications, providing complementary validation to DC resistance measurements and demonstrating appropriate electrical properties for standardized electrode testing.

## 4. Discussion

The comprehensive analysis and implementation described in this paper address fundamental accessibility barriers that currently limit widespread EEG sensing system validation. Our findings demonstrate that while diverse phantom technologies exist, each with distinct advantages and applications, a critical gap persists between the growing demand for accessible validation tools and their practical availability for resource-constrained research environments. The successful development and characterization of our 3D-printed conductive phantom bridges this gap by providing a cost-effective, accessible testing platform with appropriate electrical properties for standardized electrode evaluation.

### 4.1. Democratizing Access to EEG Validation Technologies

Current phantom technologies create significant barriers that restrict access to essential validation tools for EEG sensing system development. Commercial injection-molded phantoms, despite offering exceptional consistency and established validation credentials, create insurmountable cost barriers through their prohibitive expenses (*£*5000–20,000 tooling plus *£*300–500 per unit) and extended production timelines [13]. These barriers fundamentally limit innovation in EEG sensing technologies by preventing the majority of research groups from accessing systematic validation capabilities.

Our 3D-printed approach directly addresses these accessibility limitations by achieving an 85% cost reduction (*£*48.10 vs. *£*300–500) and fabrication time reduction from weeks to under 48 h. This dramatic improvement represents a paradigm shift toward accessible phantom technology that enables broader participation in EEG sensing system development, particularly benefiting smaller research groups, educational institutions, and laboratories in resource-constrained environments.

The approach eliminates minimum order requirements that create additional barriers in commercial phantom acquisition. The ability to fabricate single units on-demand with standard FDM printing technology enables rapid design iteration and prototyping cycles not possible with traditional manufacturing methods. This flexibility proves particularly valuable during early-stage electrode development where frequent design modifications are essential for optimization.

The democratization of phantom access particularly benefits educational institutions and developing research programs where resource constraints traditionally limited participation in EEG technology development. Students and early-career researchers gain access to validation tools previously available only to well-funded laboratories, fostering broader participation in biomedical engineering innovation and potentially accelerating the development of next-generation EEG sensing technologies.

### 4.2. Technical Performance and Validation Capabilities

The electrical characterization of our 3D-printed phantom demonstrates performance characteristics that establish its utility as a reliable testing platform for EEG sensing system validation. The measured resistance range of 821–1502 Ω provides controlled spatial variation suitable for electrode testing applications across different phantom regions, creating the heterogeneous testing environment necessary for comprehensive electrode evaluation.

Regional variations observed in our measurements, particularly the elevated resistance in Right Parietal (1502 Ω) and variable Right Occipital (851–1458 Ω) regions, create controlled testing complexity that challenges electrode performance across different phantom locations. This spatial variation provides a heterogeneous testing environment suitable for electrode positioning sensitivity and multi-channel system validation studies, enabling systematic evaluation of electrode performance under varying electrical conditions.

The phantom’s electrical properties reflect the consistent characteristics of carbon-loaded PLA material, providing reproducible testing conditions across multiple phantom units. The measured phase angles (−53° to −67°) demonstrate consistent capacitive behavior that enables controlled evaluation of electrode performance under standardized conditions. While these characteristics differ from biological tissues, which exhibit predominantly resistive behavior with phase angles of −3° to −1° at 100 Hz [91], this consistency eliminates variability that can complicate systematic electrode evaluation, providing a stable platform for comparative testing.

Signal transmission validation using 200 Hz test signals confirms bandwidth capabilities exceeding EEG requirements (0.5–100 Hz) [1] with minimal distortion characteristics, demonstrating electrical continuity appropriate for electrode testing applications. The measured amplitude retention exceeding 95% and harmonic distortion below 3% demonstrate signal fidelity suitable for electrode frequency response characterization and system validation studies.

Manufacturing repeatability analysis reveals resistance variations of less than 10% across equivalent measurement points between multiple phantom units, indicating consistency suitable for comparative studies and multi-site research collaborations. This reproducibility, combined with dimensional accuracy within ±0.1 mm tolerance, provides the standardization needed for reliable validation protocols across different research environments.

The complementary validation approach employing both DC resistance and AC impedance measurements provides comprehensive characterization of phantom performance under different electrode interface conditions. DC measurements characterize bulk material properties, while AC measurements with conductive paste simulate clinical electrode interface conditions, together providing comprehensive evaluation relevant to diverse EEG sensing applications. This multi-method approach demonstrates the phantom’s versatility for different testing scenarios and validation requirements.

### 4.3. Strategic Technology Selection Framework

The optimal phantom technology selection depends critically on specific research applications, available resources, and performance requirements. Our comprehensive analysis enables evidence-based selection decisions tailored to different research contexts and objectives, with each technology offering distinct advantages for particular applications.

For regulatory compliance testing and commercial product validation where established protocols and documentation are paramount, commercial injection-molded phantoms remain the preferred choice despite their accessibility limitations [13]. Their established validation history, documented long-term stability, and regulatory acceptance provide the necessary confidence for certification processes and quality control applications.

For preliminary electrode design evaluation and educational purposes, saline solutions offer economical and rapidly implementable alternatives [87]. Their adjustable conductivity through ionic concentration control and simple preparation make them ideal for initial testing phases and educational demonstrations where rapid iteration outweighs long-term stability requirements.

Hydrogel-based phantoms provide an intermediate solution for applications requiring controlled mechanical properties at reasonable cost (*£*30–80) [34]. These solutions offer improved durability compared to saline while maintaining accessibility, making them suitable for studies investigating electrode-tissue mechanical interactions over moderate time periods.

Research focused specifically on electrode-skin interactions and mechanical contact characteristics benefits from silicone-based phantoms, whose elasticity and surface properties provide controlled mechanical testing conditions [68]. This becomes particularly critical when evaluating dry electrodes, where mechanical pressure and conformability significantly impact signal quality [92].

Our 3D-printed approach occupies a strategically valuable position within this technology landscape, offering consistent electrical properties and controlled spatial variation while dramatically reducing cost and fabrication barriers. The demonstrated electrical compatibility with EEG systems, combined with manufacturing flexibility and accessibility, makes this approach particularly suited for iterative design processes, educational applications, and research groups with limited resources [8]. This engineering approach enables broader access to validation capabilities while supporting systematic electrode development through controlled, reproducible testing conditions.

Emerging textile-based phantoms represent an innovative approach addressing specific portability requirements [7]. The demonstrated 91.67% weight reduction compared to traditional alternatives provides unique advantages for mobile applications and field studies, though fabrication complexity may limit widespread adoption.

For advanced computational model validation requiring precise anatomical localization, multi-material phantoms offer heterogeneous electrical properties across different regions [77]. Despite their fabrication complexity and elevated cost (*£*200–600+), these sophisticated models prove essential when validating computational approaches that incorporate multiple tissue compartments or require precise spatial accuracy.

### 4.4. Economic Impact and Resource Optimization

The economic implications of phantom technology selection extend beyond initial procurement costs to encompass long-term utility, maintenance requirements, and research productivity impacts. Our cost-benefit analysis reveals significant variations in total cost of ownership across different phantom technologies, with important implications for research resource allocation and innovation capacity.

Commercial laboratories and regulatory testing facilities with substantial budgets can justify the high initial investment in injection-molded phantoms (*£*5000–20,000 tooling plus *£*300–500 per unit) through their long-term stability, reliability, and established validation credentials [5]. The amortization of tooling costs across multiple units reduces per-unit expenses for high-volume applications, making this approach economically viable for large-scale testing operations.

Academic research laboratories with moderate budgets benefit significantly from the cost-performance balance offered by our 3D-printed approach (*£*48.10 per unit). The reusability and durability of 3D-printed models provide particular long-term value, with manufacturing repeatability demonstrating less than 10% resistance variation confirming reliability needed for comparative studies over extended periods. The homogeneous conductive PLA construction enables consistent electrical properties throughout the phantom structure while maintaining manufacturing simplicity and cost-effectiveness.

Educational institutions and resource-constrained research environments experience the most significant benefits from accessible alternatives. The 85% cost reduction compared to commercial alternatives represents substantial savings that can be redirected toward instrumentation, personnel, or additional research activities. This resource reallocation can significantly enhance overall research productivity and enable broader participation in EEG sensing system development, particularly in developing countries where access to expensive validation tools has traditionally limited research capabilities.

Time-to-implementation factors significantly impact research efficiency and innovation cycles. Our 3D-printed approach, with its 47.75-hour fabrication time, provides optimal balance between accessibility and production efficiency. Projects requiring rapid prototyping benefit from the ability to implement design modifications and test results within days rather than weeks, accelerating development cycles and enabling more thorough optimization of electrode parameters.

Equipment infrastructure considerations favor laboratories with existing 3D printing capabilities, who can implement our approach with minimal additional investment. The increasing ubiquity of 3D printing technology in research environments mitigates implementation barriers, with many institutions now maintaining shared fabrication facilities accessible to multiple research teams.

### 4.5. Innovation Impact on EEG Sensing System Development

The accessibility improvements demonstrated by our 3D-printed phantom approach have transformative implications for EEG sensing system development and innovation across the research community. By removing cost and fabrication barriers, this technology enables more diverse participation in electrode development, potentially accelerating innovation in neurophysiological monitoring technologies from previously underrepresented research groups.

Small and medium enterprises developing EEG sensing technologies can now conduct comprehensive electrode testing without substantial capital investment in commercial phantoms. This accessibility reduces barriers to market entry and enables more innovative approaches to EEG sensor development, potentially leading to breakthrough technologies that might not emerge from resource-constrained environments. The rapid prototyping capabilities support iterative design methodologies essential for electrode optimization, with design modifications implementable and testable within days rather than weeks.

The open-source nature of our 3D model and documented fabrication protocols enables collaborative development and standardization across research groups. This collaborative approach can lead to improved phantom designs through community contributions and validation studies across multiple laboratories, fostering a distributed innovation model that leverages global research capabilities.

International research collaboration benefits particularly from standardized, accessible phantom technologies. Research groups in different countries can now utilize identical testing platforms, enabling meaningful comparison of results and collaborative development of electrode technologies across geographical boundaries. This standardization supports the development of global best practices and accelerates knowledge transfer between research communities.

### 4.6. Future Technological Evolution and Research Directions

Several emerging trends suggest promising directions for continued evolution in phantom head technology for EEG sensing system validation. The convergence of multiple technological advances creates opportunities for next-generation phantom designs that build upon the accessibility foundation established by our approach while addressing increasingly sophisticated validation requirements.

Advances in multi-material 3D printing technology offer increasingly sophisticated capabilities for creating heterogeneous structures with precisely controlled electrical and mechanical properties [80]. As these technologies become more accessible and cost-effective, hybrid approaches may combine the accessibility of single-material printing with controlled electrical property variation across different phantom regions, potentially creating phantoms with distinct conductivity zones while maintaining manufacturing simplicity.

The development of standardized testing protocols represents a critical evolution area [93]. Currently, the diversity of phantom technologies complicates direct comparison between studies using different validation approaches. Collaborative efforts to establish standardized benchmark phantoms and testing procedures would facilitate meaningful cross-comparison and validation of electrode technologies across research groups and institutions, building upon the standardization enabled by accessible fabrication methods.

Integration opportunities with computational modeling create possibilities for enhanced validation approaches [82]. Physical phantoms can validate computational model assumptions while computational approaches extend insights gained from physical measurements, creating comprehensive validation methodologies that leverage both approaches’ strengths.

Advanced material development promises new conductive filament formulations with tailored electrical properties, potentially enabling more precise control over phantom characteristics while maintaining manufacturing accessibility. Smart materials with responsive properties could enable dynamic phantom behavior, though such developments must balance sophistication with the accessibility advantages demonstrated in our work.

### 4.7. Broader Impact on Biomedical Engineering Education and Research

The demonstrated accessibility improvements have implications extending beyond EEG sensing system validation to broader biomedical engineering education and research capabilities. The reduced cost and fabrication barriers enable integration of phantom-based validation into undergraduate and graduate curricula, providing students with hands-on experience in biomedical device testing that was previously limited to specialized laboratories.

Educational institutions can now incorporate systematic electrode testing into biomedical engineering courses, enhancing student understanding of sensor interfaces and validation methodologies. This educational enhancement may inspire more students to pursue careers in biomedical device development and neurophysiological monitoring technologies, potentially expanding the talent pipeline for this critical healthcare technology area.

The global democratization of access enables participation from resource-constrained institutions and developing countries that previously lacked access to validation capabilities. This democratization can foster global participation in biomedical device innovation and reduce technological disparities across different regions and institutions, potentially leading to culturally appropriate and economically viable healthcare solutions for diverse global populations.

Research capacity building benefits significantly from accessible validation tools that enable smaller institutions to contribute meaningfully to electrode development research. The elimination of substantial capital investment requirements allows research resources to focus on innovation rather than infrastructure, potentially accelerating the pace of discovery and development in EEG sensing technologies.

## 5. Conclusions

This comprehensive analysis of phantom technologies for EEG sensing system validation reveals a critical gap between the growing demand for accessible validation tools and their practical availability. Current high-performance solutions create significant barriers through prohibitive costs (*£*5000–20,000 tooling plus *£*300–500 per unit) and extended production timelines that exclude the majority of research groups from systematic electrode testing. While alternative approaches offer reduced costs, they compromise on durability, consistency, or electrical stability, limiting their utility for rigorous validation studies.

Our 3D-printed conductive phantom successfully bridges this gap by demonstrating that accessible fabrication methods can deliver appropriate electrical properties for standardized EEG electrode testing. The phantom achieves an 85% cost reduction (*£*48.10 vs. *£*300–500) and fabrication time reduction from weeks to under 48 h while maintaining electrical performance suitable for comprehensive validation applications.

Systematic characterization validated the phantom’s capabilities through complementary approaches spanning DC resistance measurements (821–1502 Ω), complex impedance spectroscopy revealing controlled spatial variation (3.01–6.4 kΩ at 100 Hz with consistent capacitive behavior), and clinical EEG system compatibility testing (5–11 kΩ electrode-phantom interface impedance). The demonstrated manufacturing repeatability (resistance variations < 10%) provides the consistency required for comparative studies and multi-site research collaborations.

The phantom’s controlled electrical properties and spatial heterogeneity create a standardized testing environment that eliminates variability while providing the complexity necessary for comprehensive electrode evaluation. Clinical validation using an 8-channel EEG system confirms practical applicability for electrode development and multi-channel system validation studies, establishing the phantom’s utility across diverse EEG sensing applications.

The accessibility improvements enabled by this approach have transformative implications for EEG sensing system development. Resource-constrained laboratories, educational institutions, and small research groups gain access to validation capabilities previously limited to well-funded centers. This democratization removes barriers that have historically limited participation in electrode development, potentially accelerating innovation in neurophysiological monitoring technologies from previously underrepresented research communities.

The open-source nature of the fabrication protocols and 3D models enables collaborative development across research groups, fostering distributed innovation and knowledge sharing. Educational institutions can now integrate phantom-based validation into biomedical engineering curricula, providing students with hands-on experience in device testing and potentially expanding the talent pipeline for neurophysiological monitoring technology development.

Looking forward, this work establishes a foundation for accessible phantom technologies that can evolve with advancing 3D printing capabilities while maintaining the cost-effectiveness and manufacturing simplicity that enable widespread adoption. The demonstrated approach provides a practical framework for democratizing EEG sensing system validation, ultimately contributing to improved monitoring solutions for the 50 million people worldwide affected by epilepsy.

By proving that accessible fabrication methods can achieve performance suitable for rigorous electrode testing, this research demonstrates a pathway toward more inclusive and collaborative development of EEG sensing technologies. The systematic validation framework established here addresses critical accessibility limitations in current phantom technologies while maintaining the technical rigor necessary for meaningful validation studies, enabling broader participation in the development of next-generation neurophysiological monitoring solutions.

## Figures and Tables

**Figure 1 sensors-25-04974-f001:**
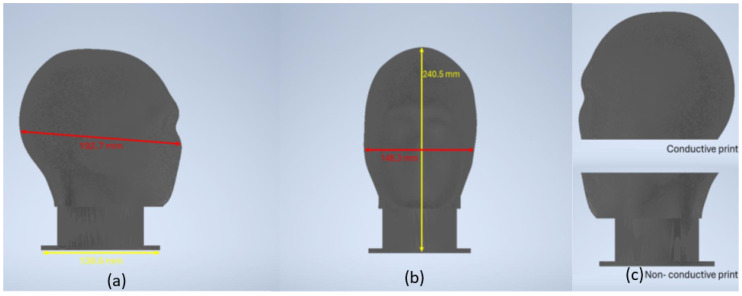
3D model of the conductive head phantom showing (**a**) side view and (**b**) front profile. (**c**) Component separation showing conductive upper section and non-conductive base.

**Figure 2 sensors-25-04974-f002:**
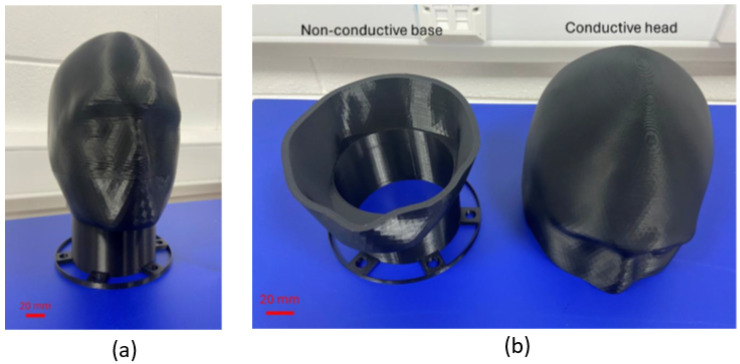
Fabricated phantom head showing (**a**) conductive head on top of non-conductive base and (**b**) components separated.

**Figure 3 sensors-25-04974-f003:**
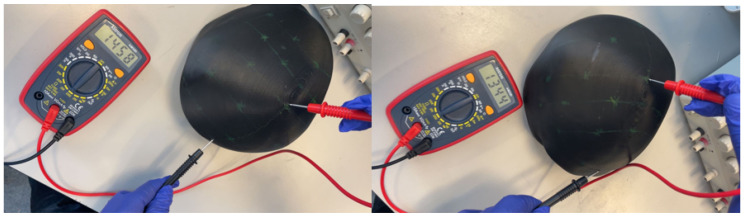
Resistance measurement setup using a digital multimeter at various anatomical locations on the conductive phantom head.

**Figure 4 sensors-25-04974-f004:**
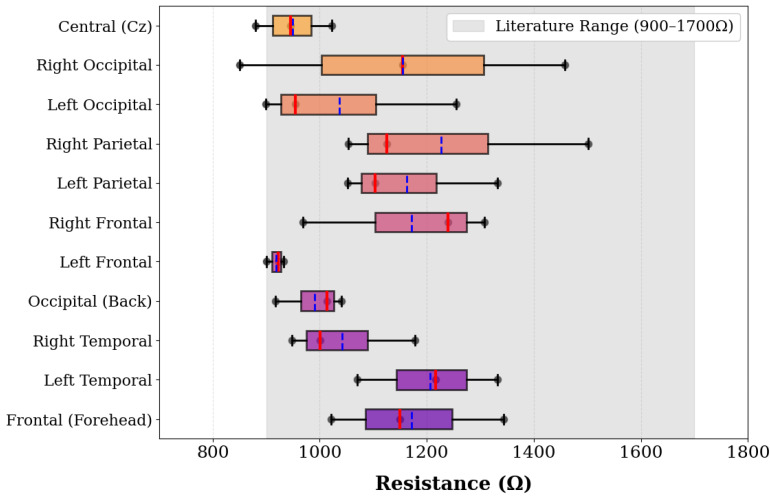
Distribution of resistance measurements across anatomical locations. Box plots represent the three measurements per location, showing median, mean, and individual measurements. The shaded region indicates the expected resistance range (900–1700 Ω) for injection-molded phantoms.

**Figure 5 sensors-25-04974-f005:**
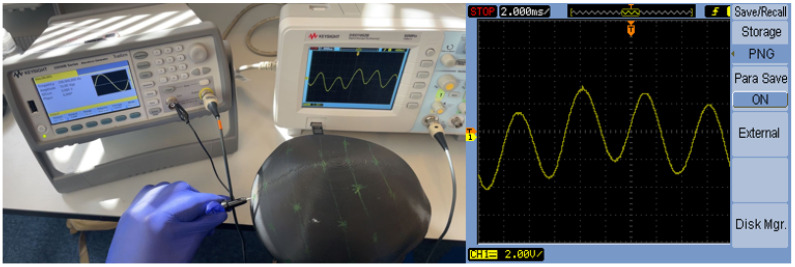
Signal transmission testing: (**left**) Experimental setup with signal generator and oscilloscope; (**right**) Output waveform showing preserved 200 Hz sine wave characteristics.

**Figure 6 sensors-25-04974-f006:**
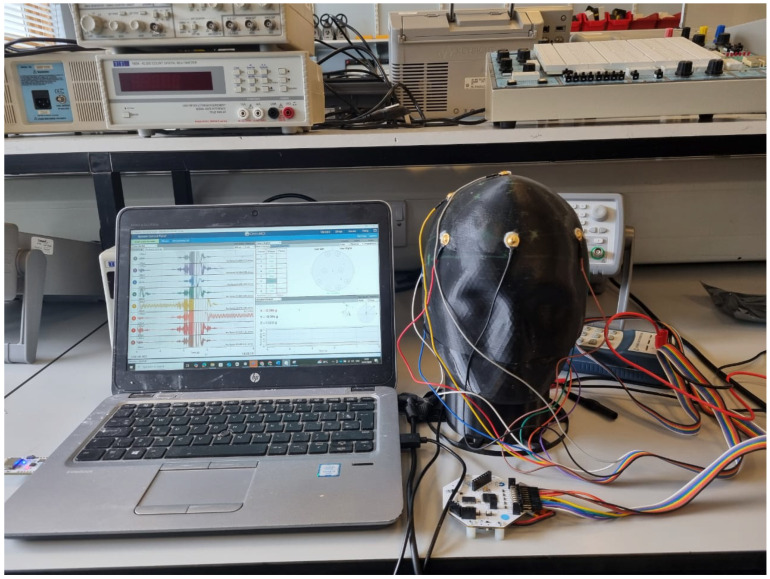
Complete OpenBCI experimental setup for phantom validation: 3D-printed conductive phantom with 8 EEG electrodes connected via multi-colored electrode wires to OpenBCI Cyton board (bottom right), with laptop computer displaying real-time EEG signal acquisition and impedance monitoring through OpenBCI GUI software (v6.0.0-beta.1).

**Figure 7 sensors-25-04974-f007:**
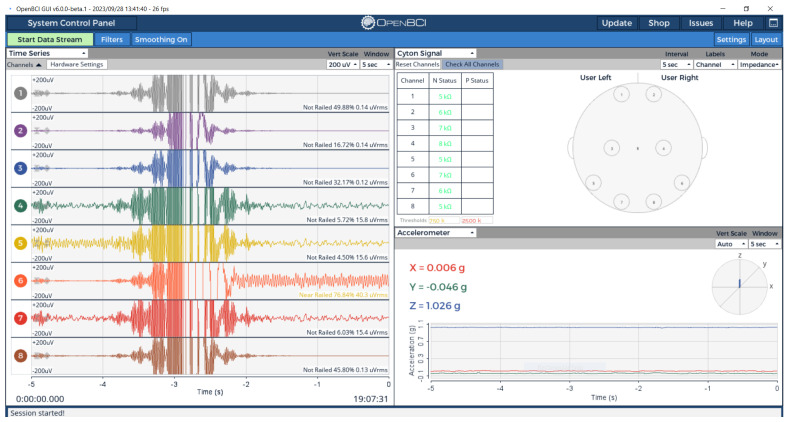
Multi-channel EEG system validation using OpenBCI Cyton: (**left**) Time-domain signal acquisition showing active signal transmission across all 8 channels; (**right**) Electrode impedance measurements demonstrating consistent values of 5–11 kΩ across connected channels, with head diagram showing electrode positions.

**Figure 8 sensors-25-04974-f008:**
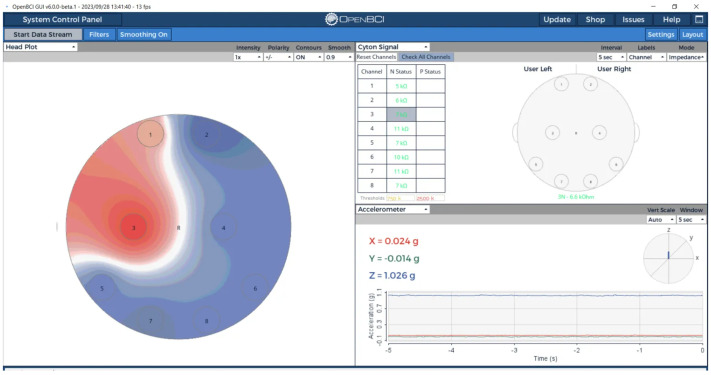
Spatial signal localization visualization using OpenBCI head plot. Active signal on Channel 3 is correctly localized (red region) with minimal crosstalk to adjacent channels, demonstrating the phantom’s capability for spatially-resolved EEG validation studies.

**Figure 9 sensors-25-04974-f009:**
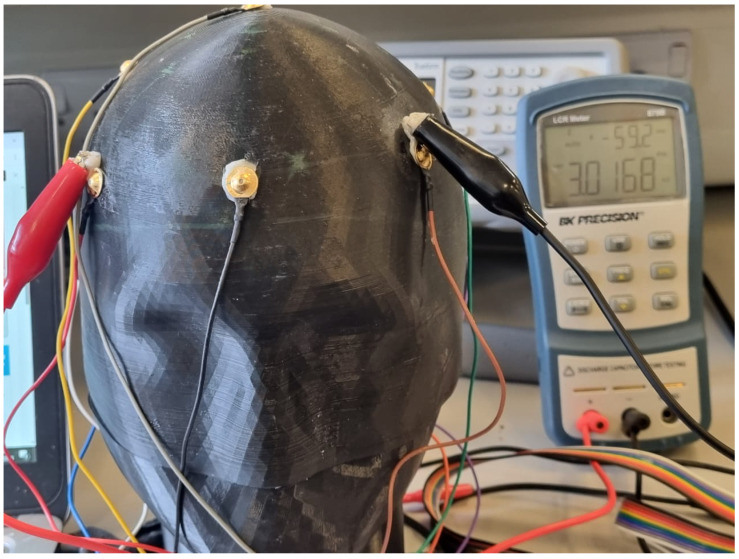
LCR meter impedance measurement setup showing probe placement on phantom frontal region. The measurement configuration demonstrates the systematic approach used for regional impedance characterization across all anatomical locations.

**Figure 10 sensors-25-04974-f010:**
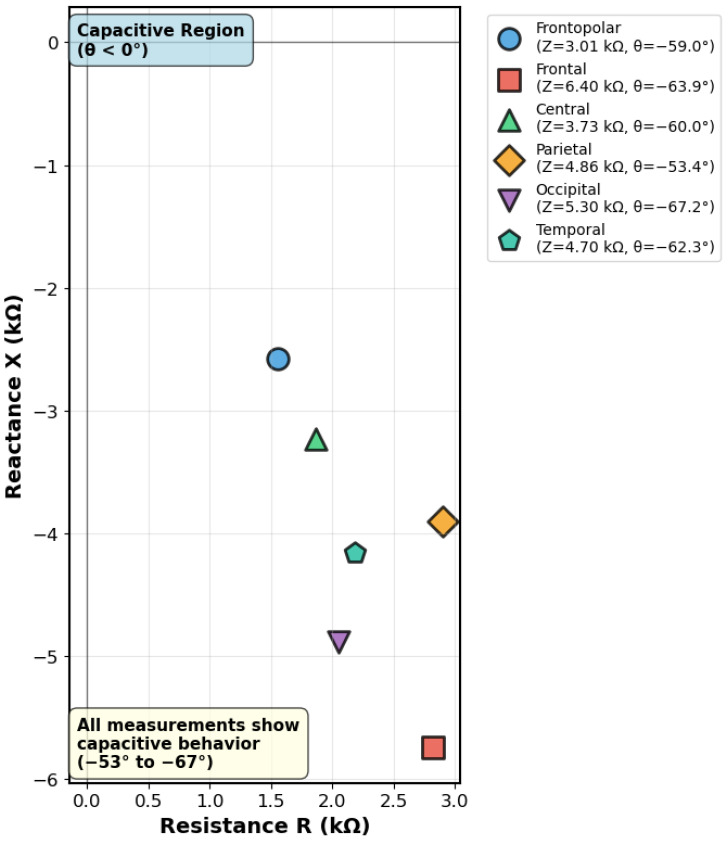
Complex impedance analysis of 3D-printed phantom across anatomical regions measured at 100 Hz using LCR meter with conductive paste. The Nyquist plot demonstrates impedance magnitude ranging from 3.01–6.4 kΩ with consistent capacitive behavior (phase angles −53° to −67°) across all regions. The spatial impedance variation provides controlled heterogeneity suitable for electrode testing while the negative reactance values confirm frequency-dependent electrical properties appropriate for EEG sensing system validation.

**Table 1 sensors-25-04974-t001:** Comparison of different phantom technologies for EEG electrode testing showing objective performance metrics.

Phantom Type	Material Cost	Equipment Cost	Production Time	Electrical Properties	Operational Lifespan
Injection-Molded [6,10]	*£*300–500/unit	*£*5k–20k tooling	Weeks	Consistent, 10–20 Ωcm	Years
Saline Solutions [17,87]	*£*10–30	Minimal	Hours	Adjustable, ionic	Days–weeks
Gelatin/Hydrogel [34,35]	*£*30–80	Minimal	1–2 days	2–5 Ωcm, adjustable	Weeks
3D-Printed [48,52]	*£*40–150	3D printer	1–3 days	15–100 Ωcm	Years [55]
Silicone-based [63,67]	*£*100–300	Molds required	2–3 days	5–15 Ωcm	Years
Multi-Material [77]	*£*200–600+	Various	3–7 days	Heterogeneous	Years
Textile-based [7]	*£*80–200	3D printer + textiles	2–3 days	1.8–2.3 kΩ	Years

**Table 2 sensors-25-04974-t002:** Optimized printing parameters for phantom fabrication.

Parameter	Conductive Section	Non-Conductive Base
Print speed [mm/s]	75	80
Printing temperature [°C]	230	225
Build plate temperature [°C]	60	60
Layer height [mm]	0.2	0.2
Line width [mm]	0.4	0.4
Infill density [%]	100	20
Infill pattern	Zig-zag	Zig-zag
Wall thickness [mm]	0.8	0.8
Support material	White breakaway	White breakaway

**Table 3 sensors-25-04974-t003:** Regional complex impedance characteristics measured at 100 Hz using LCR meter with conductive paste.

Anatomical Region	Electrode Pair	|Z| (kΩ)	θ (**°**)	R (kΩ)	X (kΩ)
Frontopolar	Fp1-Fp2	3.01	−59.0	1.55	−2.58
Frontal	F3-F4	6.40	−63.9	2.83	−5.74
Central	C3-C4	3.73	−60.0	1.87	−3.23
Parietal	P3-P4	4.86	−53.4	2.89	−3.91
Occipital	O1-O2	5.30	−67.2	2.06	−4.89
Temporal	T7-T8	4.70	−62.3	2.19	−4.16
Mean ± SD		4.67 ± 1.36	−61.0 ± 4.6	2.23 ± 0.53	−4.09 ± 1.19

## Data Availability

No new data were created or analyzed in this study. Data sharing is not applicable to this article.

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
