# Peer review of "A Cost-Effective 3D-Printed Conductive Phantom for EEG Sensing System Validation: Development, Performance Evaluation, and Comparison with State-of-the-Art Technologies"

_sensors, 2025, doi:10.3390/s25164974_

Reviewer 1 Report
Comments and Suggestions for Authors
The manuscript provides a comprehensive review of phantom head technologies for EEG electrode testing, with a particular emphasis on the development and validation of a 3D-printed conductive alternative. The study is well-structured and methodologically sound, addressing an important gap in the field by offering a cost-effective and accessible solution for EEG electrode validation. The findings are significant and have the potential to benefit researchers in resource-constrained environments. However, several areas require clarification and improvement to enhance the manuscript's impact and readability.
1. The core contributions of the study are clearly articulated, but the manuscript would benefit from a clearer distinction between the review component and the original research on the 3D-printed phantom. Consider restructuring the introduction or adding a dedicated subsection to explicitly delineate the scope of the review versus the experimental work.
2. Table 1 provides a useful overview of the technologies, but the criteria for evaluating "Accessibility" and "Durability" appear somewhat subjective. To strengthen these ratings, include quantitative metrics or references to established standards (if available).
3. The discussion of textile-based alternatives in Section 4.6 is intriguing but feels somewhat tangential. Either integrate this more thoroughly into the comparative analysis or move it to the Discussion/Future Trends section to maintain focus.
4. Section 4.1 describes the 3D printing process but lacks critical parameters (e.g., layer height, nozzle diameter) that could significantly affect conductivity and reproducibility. These details are essential to ensure replicability and should be provided.
5. The resistance measurements in Section 4.2 show considerable variability (e.g., Right Occipital: 851–1458 Ω). It would be beneficial to discuss potential sources of this variability (e.g., print quality, contact resistance) and how they were mitigated during the study.
6. The reported 85% cost reduction is compelling, but a detailed breakdown of costs (material, labor, equipment depreciation) would enhance transparency. Additionally, clarify whether the 48-hour fabrication time includes design time or only printing and post-processing steps.
7. Some related references can be added like Nano-Micro Lett. 2025, 17, 109; J. Neural Eng. 2023, 20, 026017. 
Author Response
Kindly find attached our response

Reviewer 2 Report
Comments and Suggestions for Authors
This manuscript presents a comprehensive review of phantom head technologies for EEG electrode testing and reports on the implementation of a 3D-printed conductive phantom. The long-standing challenge of balancing biological tissue mimicry and cost in phantom fabrication is a significant issue in the field, and the authors' focus on utilizing 3D printing for cost reduction while aiming for high performance addresses a relevant practical need. Based on the information provided from the Sensors Instructions for Authors, this manuscript is registered and should be reviewed as an original research article. According to the journal's guidelines, an Article is expected to report scientifically sound experiments and provide a substantial amount of new information. The quality and impact of the study are considered during peer review. While the practical motivation is appreciated, there are significant concerns regarding the manuscript's structure and the sufficiency of the experimental validation, and the amount of new information provided for an Article submission.
Here are the specific points for consideration:
- Confusion in Title and Abstract The title, which includes "A Comprehensive Review of Fabrication Methods", and the first two paragraphs of the abstract strongly suggest that this manuscript is primarily a review article. However, the latter part of the abstract describes an original implementation of a 3D-printed phantom, and the manuscript is being processed as an Article. This structure is confusing to the reader and could lead to misinterpretation of the paper's main contribution. To prevent this confusion and clearly position the manuscript as an original research article presenting a specific implementation within the context of a brief review, please revise the title and abstract accordingly.

- Validation and Scientific Soundness of Experiments The electrical characterization of the developed phantom primarily relies on resistance measurements using a digital multimeter and a signal transmission test at 200 Hz. While these measurements provide some basic electrical properties, the assessment appears insufficient for thoroughly validating a phantom intended for EEG electrode testing. EEG involves the acquisition of bioelectrical signals across a range of frequencies (typically 0.5-30 Hz). The DC resistance measured by a multimeter does not fully represent the complex impedance characteristics (including both resistance and reactance) of the phantom across this relevant frequency spectrum. A comprehensive validation of the phantom's suitability for EEG would ideally include characterization of its frequency-dependent electrical impedance. Measuring impedance magnitude and phase (e.g., using Bode plots) across the EEG frequency range, perhaps in conjunction with typical wet electrodes as used in clinical settings or even dry electrodes, would provide a more scientifically sound evaluation of its performance for neurophysiological sensing applications. For example, previous work on gelatin-based phantoms has included detailed electrical impedance spectroscopy (Owda and Casson, IEEE Acess, 2021). This would enhance the rigor of the experimental validation.

- Amount of New Information The core contribution appears to be the demonstration that a phantom suitable for EEG testing can be fabricated using commercially available conductive filament and standard 3D printing (FDM) technology at a significantly reduced cost and time compared to traditional methods like injection molding. While the cost and time reductions are indeed valuable from a practical perspective, the extent to which this demonstration provides a substantial amount of new information requires careful consideration. The fact that conductive filaments can be 3D-printed to achieve certain electrical resistance values might be anticipated based on the filament specifications and general knowledge of conductive materials. Simply demonstrating that the DC resistance falls within a certain range (821-1502 Ω) comparable to commercial phantoms might not be deemed sufficiently novel for a full Article, especially given that 3D-printed conductive phantoms are already identified as a "promising option". To increase the amount of new scientific information, the authors could explore and report on underlying technical or academic challenges in more depth. For instance, investigating the influence of using different types of conductive filaments, systematically varying the thickness or structure of the conductive layer, or analyzing the impact of printing parameters (e.g., infill density/pattern on anisotropy or layer height on conductivity variability) on the phantom's electrical properties could provide more novel insights into the fabrication process and material behavior relevant to bioelectric sensing. Addressing these points, particularly by strengthening the experimental validation and providing a more in-depth investigation into the technical aspects influencing the phantom's properties, would significantly improve the manuscript's suitability as a research article in Sensors.

Based on the significant revisions required to address the scientific soundness and novelty issues raised above, I recommend rejection and encourage resubmission by the authors.
Y. Owda and A. J. Casson, "Investigating Gelatine Based Head Phantoms for Electroencephalography Compared to Electrical and Ex Vivo Porcine Skin Models," in IEEE Access, vol. 9, pp. 96722-96738, 2021, doi: 10.1109/ACCESS.2021.3095220
Author Response
Kindly Find attached our responses to your comments.

Reviewer 3 Report
Comments and Suggestions for Authors
This paper offers significant advancements in the field of EEG electrode testing by presenting a comprehensive review of phantom technologies and introducing an innovative 3D-printed conductive solution. The study stands out for its meticulous comparison of six distinct phantom types, evaluating their electrical properties, fabrication methods, and cost-effectiveness, which provides researchers with a practical roadmap for technology selection. The development of a dual-component 3D-printed phantom using conductive PLA demonstrates remarkable resource efficiency, achieving 85% cost reduction and 48-hour fabrication while maintaining electrical performance comparable to commercial standards. The work also highlights emerging trends like textile-based alternatives, showcasing a forward-looking approach to democratizing EEG testing capabilities, especially for resource-constrained environments. By bridging the gap between affordability and technical accuracy, this research significantly contributes to advancing neurophysiological sensing applications and epilepsy monitoring solutions.
Comments:
- Please incorporate systematic validation against in vivo human EEG data, extending beyond static resistance measurements to include real-time signal comparisons in clinical settings. This would better validate the phantom’s ability to replicate neurophysiological complexities.
 - For the emerging textile-based phantoms, scenario-based comparisons with 3D printing solutions should be conducted, including aspects such as portability and anti-interference capability.

Author Response
Kindly find attached our responses to your comments,

Round 2
Reviewer 2 Report
Comments and Suggestions for Authors
The paper's refined focus on phantom fabrication has significantly improved its readability. However, from a technical perspective, several inconsistencies remain. 
1. Apparent Incomparability of DC Resistance (R) and 100 Hz Impedance (|Z|) in Table 3, despite its Central Role in the Manuscript's Claims.
The comparison of DC resistance (R) and 100 Hz impedance (|Z|) in Table 3 appears problematic, despite being central to the manuscript's claims. These values seem incomparable because DC measurements were conducted with direct probe contact, while 100 Hz AC measurements used conductive paste. This difference in methodology can influence frequency characteristics. Furthermore, biological tissues typically exhibit higher DC resistance than 100 Hz impedance. However, the phantom's DC resistance (821-1502 Ω) is reported as significantly lower than its 100 Hz impedance (|Z| 3.01-6.4 kΩ, R 1.55-2.89 kΩ), which contradicts this general physiological characteristic. The paper's claim that the phantom possesses "frequency-dependent" and "realistic tissue-like" electrical properties based on this comparison may be questionable.
2. Lack of Comparison of the 100 Hz Phase Angle (~-65°) with Actual Skin Characteristics.
The manuscript asserts that the 100 Hz phase angle of approximately -65° (-53.4° to -67.2°) is suitable for EEG sensing system validation. However, it lacks comparison with actual skin properties. Previous research [1] on porcine skin for EEG dry electrodes shows a 100 Hz phase angle of only -3° (Day 1) to -1° (Day 7). The phantom's significantly larger capacitive behavior compared to real skin is not quantitatively evaluated to determine its impact on EEG system validation. Therefore, claiming accurate mimicry of realistic biological tissue solely based on the presence of capacitive characteristics seems insufficient.
[1] A. Y. Owda and A. J. Casson, "Investigating Gelatine Based Head Phantoms for Electroencephalography Compared to Electrical and Ex Vivo Porcine Skin Models," in IEEE Access, vol. 9, pp. 96722-96738, 2021, doi: 10.1109/ACCESS.2021.3095220.
Author Response
Kindly find attached our responses.
